# Degradation Behaviour of Mg0.6Ca and Mg0.6Ca2Ag Alloys with Bioactive Plasma Electrolytic Oxidation Coatings

**Lara Moreno** [1,*] **, Marta Mohedano** [1] **, Beatriz Mingo** [2] **, Raul Arrabal** [1] **and Endzhe Matykina** [1]

1   Departamento de Ingeniería Química y de Materiales, Facultad de Ciencias Químicas, Universidad Complutense, 28040 Madrid, Spain; mmohedan@ucm.es (M.M.); raularrabal@quim.ucm.es (R.A.); ematykin@ucm.es (E.M.)
2   School of Materials, The University of Manchester, Oxford Road, Manchester M13 9PL, UK; beatriz.mingo@manchester.ac.uk
*   Correspondence: laramo01@ucm.es

**Abstract:** Bioactive Plasma Electrolytic Oxidation (PEO) coatings enriched in Ca, P and F were developed on Mg0.6Ca and Mg0.6Ca2Ag alloys with the aim to impede their fast degradation rate. Different characterization techniques (SEM, TEM, EDX, SKPFM, XRD) were used to analyze the surface characteristics and chemical composition of the bulk and/or coated materials. The corrosion behaviour was evaluated using hydrogen evolution measurements in Simulated Body Fluid (SBF) at 37 °C for up to 60 days of immersion. PEO-coated Mg0.6Ca showed a 2–3-fold improved corrosion resistance compared with the bulk alloy, which was more relevant to the initial 4 weeks of the degradation process. In the case of the Mg0.6Ag2Ag alloy, the obtained corrosion rates were very high for both non-coated and PEO-coated specimens, which would compromise their application as resorbable implants. The amount of F⁻ ions released from PEO-coated Mg0.6Ca during 24 h of immersion in 0.9% NaCl was also measured due to the importance of F⁻ in antibacterial processes, yielding 33.7 μg/cm², which is well within the daily recommended limit of F⁻ consumption.

**Keywords:** corrosion; PEO; biodegradable implants

## 1. Introduction

Over the last decades, magnesium and its alloys have been extensively investigated due to their excellent properties that make them promising candidates for biodegradable orthopaedic implants and cardiovascular stents [1]. For instance, the fracture toughness of magnesium is greater than that of ceramic biomaterials and its elastic modulus and compressive yield strength are closer to those of natural bone [2,3]. Moreover, Mg is an essential metallic element in the human body with several functional roles and is present in the bone tissue; for instance, magnesium has a stimulatory effect on the growth of new bone tissue [4,5].

The main handicap of magnesium and its alloys is the low corrosion resistance that causes a reduction in the mechanical integrity of the implant. In order to overcome this limitation, different strategies have been reported and among them alloying elements and surface modifications stand out [6,7].

The alloying design strategy has been studied in depth in the case of magnesium alloys for structural applications and has been mainly focused on commercial Mg-Al-Zn (AZ) alloys [8] due to the beneficial effect of aluminium and zinc on castability, corrosion and mechanical properties [9,10]. On the contrary, the presence of elements such as Fe and Ni decrease the corrosion properties due to formation

of micro-galvanic couples [11]. However, in those applications the toxicity and biocompatibility of the alloys are not taken into account [12]. In the biomedical filed, where biocompatibility is the main requirement, different elements have been studied following criteria of non-toxicity and absorbability in the human body [12–14].

Among them, calcium (Ca) and silver (Ag) present a great interest: (i) calcium is the main bone component, is an essential element in the human body and participates in cell signaling reactions [5,15]; (ii) silver is an element in the human body that has excellent antibacterial activity [16,17], it is also effective in the treatment of some microbes.

In the particular case of Mg alloys containing Ca and/or Ag, only a few works have been reported. For instance, the corrosion performance of MgXCa (X = 0.5–10 wt.% [18,19]) was studied and it was concluded that a low amount of Ca (up to 1%) decreases the corrosion rate, leads to bone regeneration around the implant and does not induce cytotoxicity. From the mechanical properties point of view, alloying with Ca (<4%) has been reported to increase the tensile strength [20,21].

Regarding magnesium alloys containing Ag, their antibacterial effect has been reported for a wide range of the microbial spectrum [22]. However, problems of high corrosion rates [16] and cytotoxicity [23] have also been found, although the cytotoxicity can be decreased while maintaining the antimicrobial effect if a secondary element, such as calcium, is included in the alloy [19].

Regarding the second strategy to improve the corrosion resistance based on surface modification of Mg and its alloys, the Plasma Electrolytic Oxidation (PEO) technique is one of the most promising candidates, as modified surfaces show improved corrosion behaviour along with other properties (e.g., biocompatibility) [24]. PEO generates anticorrosive, biocompatible and bioactive ceramic coatings with tailored composition, roughness, microstructure and porosity that can be controlled by optimization of the process electrical parameters and the composition of the electrolyte [24–26].

With respect to the PEO of Mg–Ca alloys, there are a few studies that reported an improved corrosion resistance [13,24,27,28], cell adhesion and bone regeneration [26,29,30]. However, to date there are no studies on PEO coatings on MgCaAg alloys.

The present work compares the corrosion resistance of Mg0.6Ca and Mg0.6Ca2Ag alloys in SBF with and without surface modification in order to determine their suitability for biodegradable implants.

## 2. Materials and Methods

Cast ingots of Mg0.6Ca and Mg0.6Ca2.0Ag (composition shown in Table 1) alloys were supplied by Magnesium Innovation Centre (MagIC, Helmholtz-Zentrum Geesthacht, Geesthacht, Germany). The ingots were cut into $60 \times 6 \times 4$ mm$^3$ bars and ground on all sides to P1200 through successive grades of SiC abrasive paper, rinsed in isopropyl alcohol and dried in warm air. Electrical contact was provided at one end of the bars through a 1.5 mm metric threaded hole.

**Table 1.** Spark OES analysis of Mg0.6Ca and Mg0.6Ca2Ag alloy composition (wt.%).

| Element | Mg0.6Ca | Mg0.6Ca2Ag | Element | Mg0.6Ca | Mg0.6Ca2Ag |
|---------|---------|------------|---------|---------|------------|
| Mg | 99.4 | 97 | Nd | 0.00107 | 0.0009 |
| Ca | >0.504 | >0.504 | Si | <0.0010 | <0.0010 |
| Ag | 0.0005 | 2.417 | Sr | 0.0006 | 0.0007 |
| Pr | 0.02844 | 0.02579 | Sn | <0.0005 | <0.0005 |
| Al | 0.01985 | <0.00020 | Ni | 0.0004 | 0.0005 |
| Th | 0.01547 | 0.01437 | P | <0.0003 | <0.0003 |
| Mn | 0.015 | 0.03715 | Zr | <0.0003 | <0.0003 |
| Cu | 0.00156 | 0.0004 | Fe | <0.0002 | <0.0002 |

PEO treatments were conducted using alternating current (AC) voltage-controlled EAC-S2000 (ET Systems electronic Gmbh, Altlußheim, Germany) power supply in a 2 L double jacket thermostated glass cell using a stainless steel mesh (AISI 316 of Ø15cm) as a counter electrode. The coatings were developed in a Ca–P containing electrolyte solution (10 g/L Na$_2$PO$_4$, 1 g/L KOH, 8 g/L NaF and 2.9 g/L

CaO) using a square voltage input waveform with positive and negative amplitudes of 430 and 50 V, respectively, 50% duty cycle, a frequency of 50 Hz and an rms current limitation of 138 mA/cm$^2$.

The alloy microstructure and coating composition and morphology were examined by scanning electron microscopy (JEOL JSM-6400, Tokyo, Japan) equipped with an energy dispersive X-ray (EDS) microanalysis system (OXFORD LINK PENTAFET 6506, Abingdon, Oxfordshire, UK). Metallographic preparation of the specimens included grinding from P120 to P1200 grit of SiC abrasive papers and polishing to 1 μm using diamond paste.

The alloys were also examined by transmission electron microscopy (TEM) using a JEM 2100HT JEOL instrument equipped with EDS operated at 200 keV of acceleration voltage. TEM specimens were prepared as 3 mm diameter and 0.1 μm-thick disks thinned until perforation by ion milling.

Surface potential maps of the alloys were obtained using a NANOSCOPE IIIA Multimode Scanning Kelvin Probe Force Microscope (SKPFM, Bruker, Billerica, MA, USA) with Pt coated Si tip operated in tapping mode. Topographic images and potential maps were obtained with the tip-to-sample distance fixed at 100 nm. The tip to sample distance was kept constant at 100 nm using a two-pass technique, where the height data is recorded in tapping mode during the first pass and the tip lifts above the surface to an adjustable lift height and scans the same line while following the height profile recorded in the second pass. All measurements were made at room temperature with a relative humidity in the range of 40%–65%.

X-ray diffraction (PHILIPS XPERT instrument, Amsterdam, The Netherlands) was used to characterize the phase components of the alloys and coatings using Cu-K$\alpha$ = 1.54056 Å radiation and 2θ values between 10° and 90° at steps of 0.04°/2 s. A PANanalytical Aeris Research Edition was used to characterize Mg0.6ca2Ag using 2θ values between 10° and 30° at steps 0.04°/298.60 s. The spectra was analysed with the X-Pert High Score Plus program.

Hydrogen evolution measurements were carried out during immersion of PEO-coated and non-coated bars in m-SBF (5.403 g/L NaCl, 0.504 g/L NaHCO$_3$, 0.426 g/L Na$_2$CO$_3$, 0.225 g/L KCl, 0.230 g/L K$_2$HPO$_4$·3H$_2$O, 0.311 g/L MgCl$_2$·6H$_2$O, 17.892 g/L HEPES, 0.293 g/L CaCl$_2$, 0.072 g/L Na$_2$SO$_4$) [31] solution at 37 °C for up to 65 days using 20 mL/cm$^2$ of solution. The m-SBF was changed every 48 h or as needed if pH > 8.4. The electrical contact holes in the bars were sealed using a Lacquer 45 Stopping-off Resin (MacDermid Plc, Birmingham, UK).

Fluoride ions released during immersion of PEO-coated Mg0.6Ca alloy specimens (10.58 cm$^2$ of surface area) in 0.9% NaCl at 37 °C were measured using a fluoride ions selective electrode (Crison, Barcelona, Spain), which consisted of a lanthanum fluoride monocrystal membrane doped with europium. The membrane potential difference depended on the concentration of F$^-$ in the solution and was measured using a reference electrode of Ag/AgCl and pH meter GLP22 (Crison) every 30 min during first 2 h of immersion and at 24 h. The calibration curve was carried out using fluoride reference solutions (500 ppb, 1, 2.5, 5 and 10 ppm) prepared from a fluoride standard of 100 μg/L. Each reference solution was prepared using 25 mL of a TISAB solution (58 g/L NaCl, 4 g/L C$_{14}$H$_{22}$N$_2$O$_8$ in distilled water, pH adjusted to 5.0–5.5 with 6 M NaOH) in order to normalize the ionic strength.

## 3. Results

### 3.1. Characterisation

For both materials, dendritic $\alpha$-Mg grains were revealed (Figure 1) with the second phase distributed along the grain boundaries in the form of a discontinuous network and decorated the interdendritic regions (bright contrast regions).

The EDS analysis of Areas 1 and 4 correspond to the matrix of Mg0.6Ca and Mg0.6Ca2Ag respectively (Table 2). Points 2 and 3 (Figure 1b) were located in the second phase of the Mg0.6Ca and revealed mainly a similar amount of Mg and Ca elements. In the case of the Mg0.6Ca2Ag alloy, two different EDS analyses (Figure 1d, Points 5 and 6) showed the formation of particles with different contents of Ag.

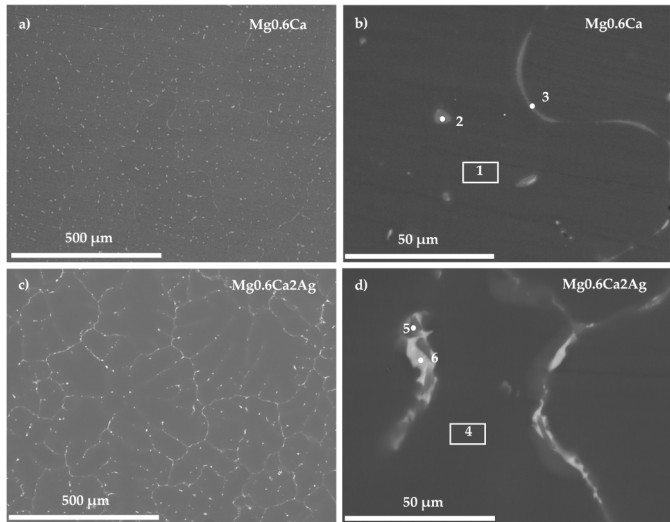

**Figure 1.** Backscattered electron micrographs of the studied alloys: (**a**,**b**) Mg0.6Ca and (**c**,**d**) Mg0.6Ca2Ag.

**Table 2.** Local EDS point analysis as per Figure 1.

| Alloy | Location | Mg | Ca | Al | Si | Ag |
|---|---|---|---|---|---|---|
| Mg0.6Ca | 1 | 96.77 | 2.3 | 0.73 | 0.2 | – |
| | 2 | 90.2 | 9.42 | 0.38 | – | – |
| | 3 | 92.66 | 7.11 | 0.27 | – | – |
| Mg0.6Ca2Ag | 4 | 99.4 | 0.15 | – | – | 0.4 |
| | 5 | 98.13 | 0.3 | – | – | 1.57 |
| | 6 | 76.03 | 7.47 | – | – | 15.57 |

TEM analyses of Mg0.6Ca alloy displayed the second phase (Figure 2a) as an eutectic aggregate with lamellar morphology, probably formed by $\alpha$-Mg/Mg$_2$Ca phases in accordance with the EDS analysis (Table 3), the Mg–Ca phase diagram and other works on binary Mg–Ca alloys [25,32–34]. In addition, polygonal shape particles (Figure 2b) containing impurities were found (Point 2, Table 3).

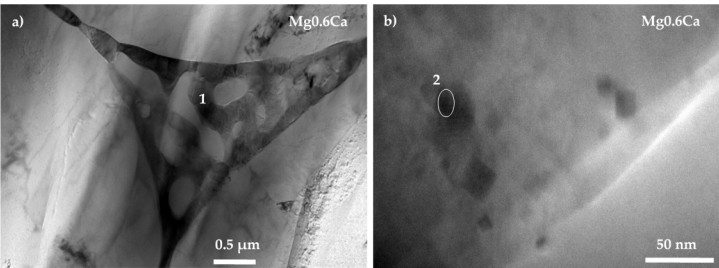

**Figure 2.** Transmission electron micrographs of the Mg0.6Ca alloy with locations of EDS analysis: (**a**) Mg0.6Ca 0.5 μm and (**b**) Mg0.6Ca 0.5 nm.

**Table 3.** Local EDS analysis (at.%) of intermetallic particles as per Figures 2 and 3.

| Alloy | Location | Mg | Ca | Ag | Fe | Al | Ni | Co | Cr | Si |
|---|---|---|---|---|---|---|---|---|---|---|
| Mg0.6Ca | 1 | 92.59 | 7.08 | – | – | 0.33 | – | – | – | – |
| | 2 | 85.04 | 12.7 | – | 0.05 | 1.29 | 0.86 | – | – | – |
| Mg0.6Ca2Ag | 3 | 74.6 | 7.27 | 16.52 | 0.59 | – | 0.15 | 0.51 | 0.62 | 0.15 |
| | 4 | 99.08 | 0.15 | 0.31 | 0.15 | – | – | 0.12 | 0.18 | – |

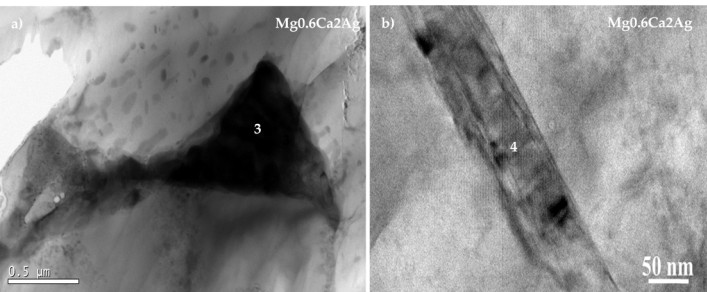

**Figure 3.** Transmission electron micrographs of the Mg0.6Ca2Ag alloy with locations of EDS analysis. (**a**) Mg0.6Ca2Ag 0.5 μm and (**b**) Mg0.6Ca2Ag 50 nm.

In the case of the ternary alloy (Mg0.6Ca2Ag) (Figure 3), TEM studies were not conclusive as the zones of the ion milled lamella where particles were located were too thick for generating an informative electron diffraction pattern.

In fact, the available data of TEM studies on Mg alloys containing silver are very scarce and all of them have been conducted on binary Mg–Ag alloys reporting the formation of different intermetallic compounds such as MgAg, MgAg$_3$ or MgAg$_4$ [35,36]. The present work revealed the formation of particles with different shapes (Figure 3) and compositions (Table 4), some of them were Ag-enriched (Table 3, Point 2).

**Table 4.** EDS analysis (at.%) of Mg0.6Ca alloys at the locations of SKPFM mapping as per Figure 4.

| Alloy | Location | Mg | Ca | Si | O | Al |
|---|---|---|---|---|---|---|
| | 1 | 91.31 | 6.15 | – | 2.23 | 0.3 |
| Mg0.6Ca | 2 | 80.42 | 7.38 | 10.87 | 1.33 | – |
| | 3 | 95.67 | 4.33 | – | – | – |

The local Volta potential difference (VPD) between constituents on a submicron scale was obtained using SKPFM. Surface potential maps and potential profile along with the SEM images of the studied area are displayed in Figure 4 for the Mg0.6Ca and Mg0.6Ca2Ag alloys, respectively.

Figure 4a,d show the selected region of the Mg0.6Ca alloy where surface potential maps and profiles were acquired. Point 1 corresponds to the Mg$_2$Ca phase that shows a negative potential with respect to the α-Mg matrix (VPD of ~−50 mV). It is important to note that the available data regarding the electrochemical behaviour of Mg$_2$Ca is quite limited and controversial results are reported about the anodic [37,38] as well as cathodic behaviour [19] of this phase, which have been mainly attributed to the differences in the composition of the alloy (e.g., presence of impurities) and matrix segregation. Points 2 and 3 (Figure 4d) correspond to an impurity (Point 2) and to the Mg$_2$Ca phase (Point 3, Table 4). As it was observed before, the latter shows an anodic behaviour (VPD ~−15 mV) compared to the matrix, whereas the presence of impurities reveals a slight cathodic behaviour with respect to the α-Mg matrix (VPD ~+20 mV) due to the presence of more noble elements such as Si, Fe or Al.

For the Mg0.6Ca2Ag alloy, the Volta potential profiles conducted in the inclusion (Figure 5) reveal a different electrochemical response depending on the elemental composition analyzed by EDS (Table 5). The area enriched in Ag (Ca/Ag ratio of 0.89) (Figure 5b) revealed a cathodic behaviour (VPD ~+20 mV) in comparison with the α-Mg matrix, whereas the Ca/Ag ratio of 2.39 leads to an anodic performance with potential differences around VPD ~−27 mV. Ben et al. studied the electrochemical behaviour of magnesium–silver ternary alloys (MgZnAg) and reported a cathodic behaviour of MgAg secondary phase [39], which appears to be similar to the behaviour of the areas enriched with Ag in Mg0.6Ca2Ag alloy of the present work.

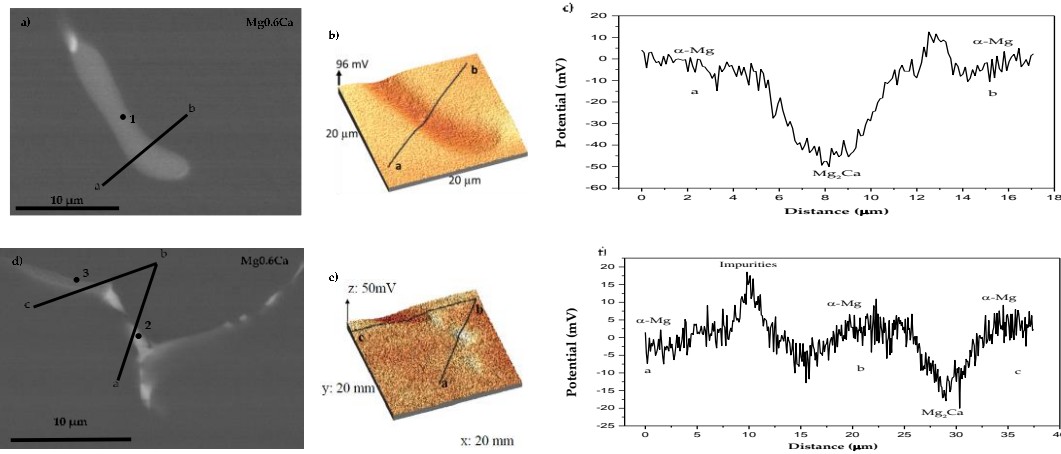

**Figure 4.** SEM backscattered images (**a**,**d**), surface potential maps (**b**,**e**) and potential profile (**c**,**f**) in selected areas of the Mg0.6Ca alloy.

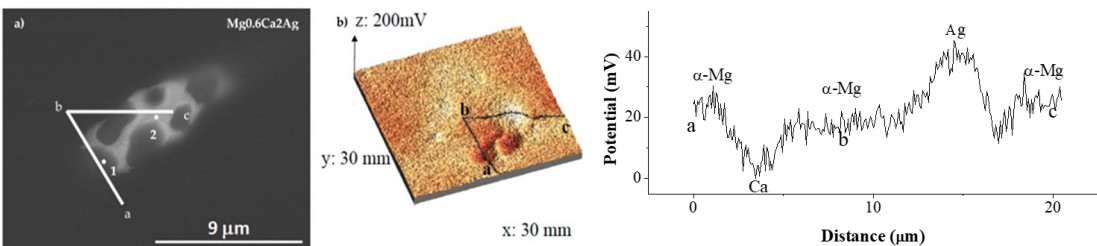

**Figure 5.** SEM backscattered image (**a**), surface potential maps (**b**) and potential profile (**c**) in selected areas of the Mg0.6Ca2Ag alloy.

**Table 5.** EDS analysis (at.%) of the Mg0.6Ca2Ag alloy at the locations of SKPFM mapping as per Figure 5.

| Alloy | Location | Mg | Ca | O | Ag |
|---|---|---|---|---|---|
| Mg0.6Ca2Ag | 1 | 91.44 | 5.04 | 1.42 | 2.11 |
| | 2 | 81.67 | 8.32 | 0.71 | 9.31 |

Figure 6 shows the plan view and cross-sectional coating morphologies of Mg0.6Ca/PEO and Mg0.6Ca2Ag/PEO coatings. In both alloys the coating surface presents a typical crater-like porous morphology associated with the sites of discharge channels, gas evolution and rapid solidification phenomena [24,40]. In both Mg0.6Ca/PEO and Mg0.6Ca2Ag/PEO coatings the surface Ca/P ratio is relatively high (1.42 and 1.53, respectively) compared with the inner regions of the coating (Table 6), although lower than that of biological hydroxyapatite (1.67) [41], and both Ca and P contents increase towards the coating/electrolyte interface. The surface enrichment of biomaterials (e.g., of Ti alloys) in Ca and P is well known to improve the initial cell response [42].

The cross-sectional images (Figure 6c,d) reveal relatively uniform coatings with thicknesses in the range of 32–35 μm. It can be observed that both coatings are constituted by three layers. A thin barrier layer (less than 1 μm) adjacent to the substrate is mainly composed of Mg, O and F and Ag (the latter only in the case of MgCaAg, Table 6). An intermediate region with small pores constitutes ~40% of the coating thickness. An outer, more compact, region contains a few but relatively large pores.

It is worth mentioning that EDS analysis for the MgCaAg alloy did not detect the presence of Ag in the coating surface, whereas the Ag content in the inner regions did not exceed ~0.2 at.%, suggesting that a rather limiting if any antibacterial effect can be expected at the initial stage of the implantation.

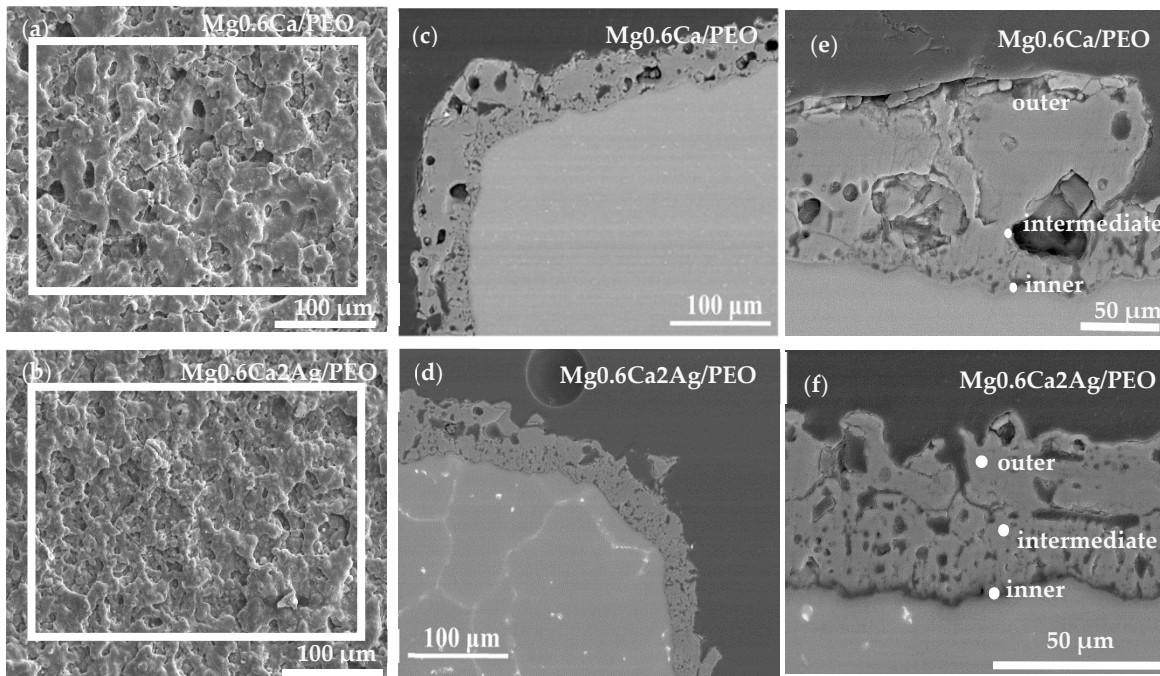

**Figure 6.** Backscattered electron micrographs of coating surface morphologies and cross-sections after corrosion of Mg0.6Ca/PEO (**a**,**c**,**e**) and PEO-Mg0.6Ca2Ag (**b**,**d**,**f**).

**Table 6.** Compositions of Mg0.6Ca/PEO and Mg0.6Ca2Ag/PEO before corrosion by EDX (at.%).

| Alloy | Location | Mg | O | F | Na | P | Ca | Ag | Ca/P |
|---|---|---|---|---|---|---|---|---|---|
| Mg0.6Ca | Surface | 22.1 | 44.9 | 16.3 | 4.2 | 5.2 | 7.3 | – | 1.42 |
| | Inner layer | 39.7 | 24.5 | 32.5 | 0.3 | 2.9 | 0.1 | – | 0.63 |
| | Intermediate layer | 35.2 | 21.2 | 36.7 | 1.9 | 4.4 | 0.6 | – | 0.14 |
| | Outer layer | 36.5 | 34.9 | 18.6 | 4.2 | 3.6 | 2.3 | – | 0.05 |
| Mg0.6Ca2Ag | Surface | 27 | 42.5 | 16.2 | 3.7 | 4.2 | 6.4 | – | 1.53 |
| | Inner layer | 37.6 | 22.3 | 34.7 | 1 | 3.8 | 0.5 | 0.2 | 0.4 |
| | Intermediate layer | 32.4 | 26.3 | 30.1 | 4.3 | 5.1 | 1.5 | 0.2 | 0.3 |
| | Outer layer | 36.2 | 37.3 | 13 | 4.6 | 6.3 | 2.5 | – | 0.12 |

The XRD analyses of uncoated materials revealed peaks corresponding to α-Mg for both alloys (Figure 7). In the case of uncoated Mg0.6Ca, no intermetallic/secondary phases were detected probably due to their negligible amount. Although, in our previous study of a similar alloy (cast Mg0.8Ca) a formation of Mg$_2$Ca phase was confirmed [25]. Most of the diffraction peaks of Mg0.6Ca2Ag were the same as those for Mg0.6Ca except that there were small peaks in a low 2θ angles region corresponding to binary and ternary intermetallic phases as Mg$_2$Ca, MgAg$_4$, Mg$_{54}$Ag$_{17}$ [25,36,43] and Ca$_2$MgAg$_3$ (Figure 7, inset). A further systematic TEM and electron diffraction study would be necessary in order to confirm the presence of these phases.

On the other hand, PEO coatings revealed peaks corresponding to the substrate, and the formation of crystalline phases such as MgO, MgF$_2$, CaF$_2$ and Ca$_5$(PO$_4$)3F was detected. The MgO is formed due to electrolytic oxidation of the substrate and the other phases are formed due to plasma-chemical reactions between the ions of the electrolyte and the substrate inside the microdischarge channels [44].

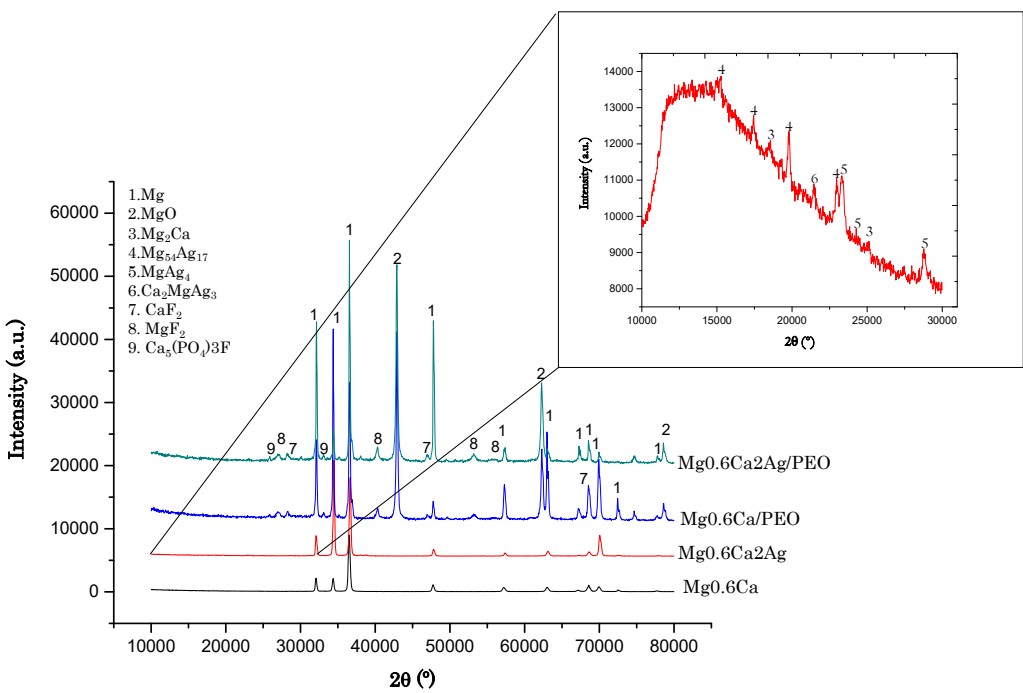

**Figure 7.** XDR patterns from bulk material and PEO coatings.

### 3.2. Hydrogen Evolution Measurement

Figure 8a,b show the hydrogen evolution volume and hydrogen evolution rate for Mg0.6Ca and Mg0.6Ca/PEO after 60 days of immersion in SBF at 37 °C. Figure 9 shows the volume of evolved hydrogen for Mg0.6Ca2Ag and Mg0.6Ca2Ag/PEO after 4 days of immersion. As expected, the uncoated material initially exhibited a high amount of hydrogen with an evolution rate of 3.86 mL/cm$^2$ week after 1 week of immersion, with progressive decrease to 1.02 and 1.11 mL/cm$^2$ week after 6 and 8 weeks of immersion, respectively. The decrease of hydrogen evolution rate is due to the generation of a corrosion products layer, which acts as a partially protective coating.

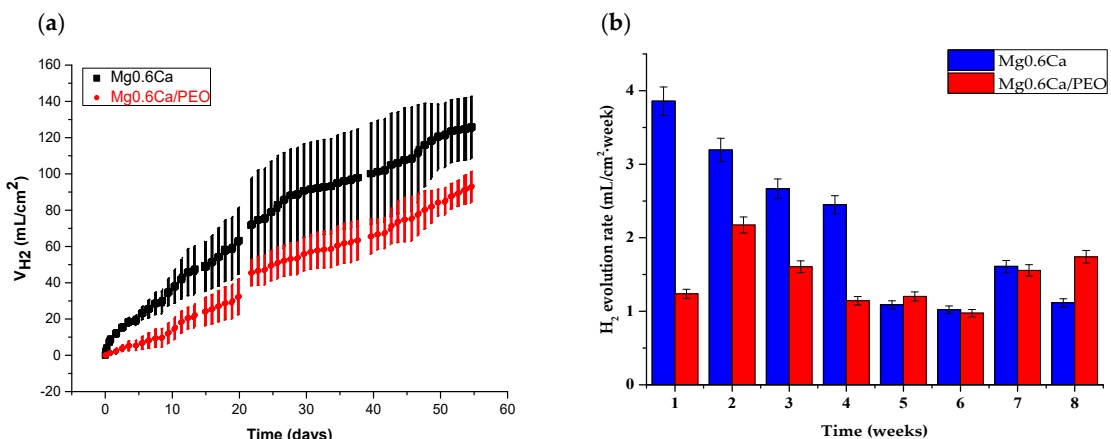

**Figure 8.** (**a**) Hydrogen volume and (**b**) hydrogen evolution rate of Mg0.6Ca and Mg0.6Ca/PEO after 60 days of immersion in m-SBF.

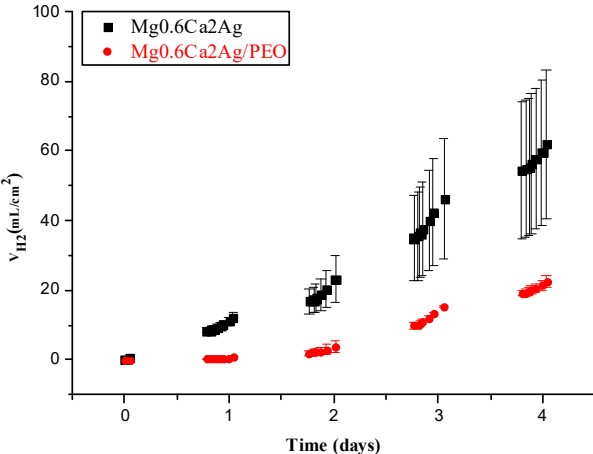

**Figure 9.** Hydrogen volume for Mg0.6Ca2Ag and Mg0.6Ca2Ag/PEO after 4 days of immersion in m-SBF.

Mg0.6Ca/PEO showed a considerably reduced hydrogen evolution rate during the first 4 weeks (~1.11 mL/cm$^2$ week); however, after that time both non-coated and PEO-coated materials reached a similar degradation rate. Further increase of the hydrogen evolution rate was evident after 6 weeks (1.75 mL/cm$^2$ week) due to the loss of protective properties of the inner PEO layer and increased electrochemical activities in the substrate/coating interface [45–47]. The latter degradation rate corresponds to 1.9 mg/(cm$^2$ week) of mass loss or 11 μm/week of thickness loss.

Both Mg0.6Ca2Ag and Mg0.6Ca2Ag/PEO exhibited an extremely high degradation rate (Figure 9), corresponding to ~60 or ~20 mL/cm$^2$, respectively, and the experiments were stopped after 4 days of immersion.

Figure 10 shows the macro degradation photos of the specimens. Both coated and non-coated Mg0.6Ca alloy (Figure 10a,b) presented a uniform corrosion with similar loss of material and dimensions. Mg0.6Ca2Ag also showed a generalized, but heavily heterogeneous corrosion morphology for the non-coated specimens (Figure 10c) and localized corrosion for the PEO-coated ones; both were found completely disintegrated after about 10 days of immersion.

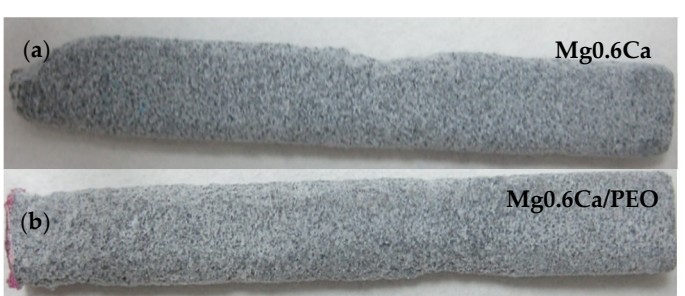
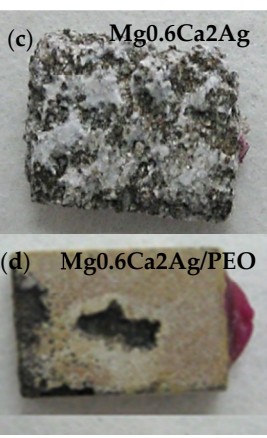

**Figure 10.** 3D macrograph of (**a**) Mg0.6Ca, (**b**) Mg0.6Ca/PEO after 60 days, (**c**) Mg0.6Ca2Ag and (**d**) Mg0.6Ca2Ag/PEO after 4 days of immersion in m-SBF.

Following the immersion, a heavy generalized corrosion was observed in all cases (Figure 11) and PEO coatings were evidently detached, and, in the case of the Mg0.6Ca2Ag/PEO system, this had already occurred by the 4th day of immersion. A complete loss of adhesion is not a typical behaviour for PEO-coated Mg alloy, as was previously demonstrated by the authors in [25], and a 40–50 μm-thick

coating can be expected to remain mostly adhered even after 8 weeks of immersion with a thick corrosion product layer developing underneath it.

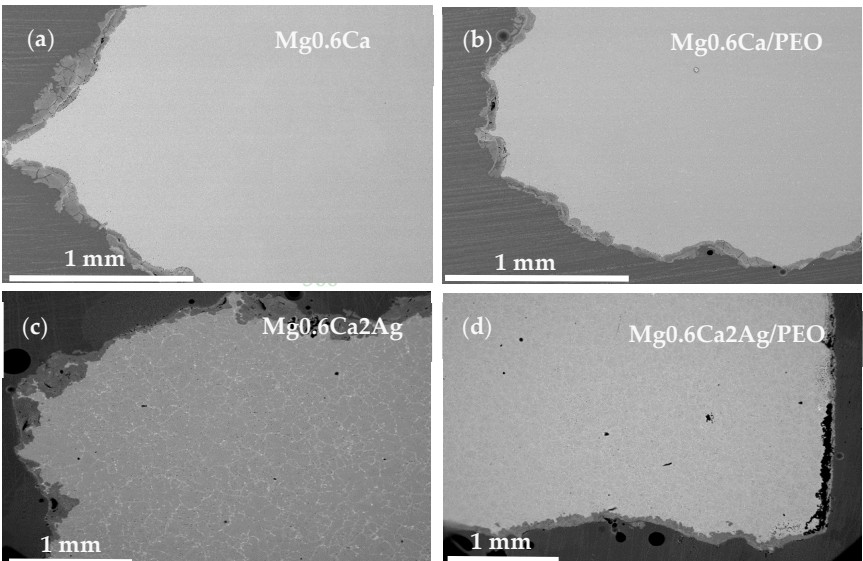

**Figure 11.** Cross-section before corrosion (**a**) Mg0.6Ca, (**b**) Mg0.6Ca/PEO after 60 days and (**c**) Mg0.6Ca2Ag and (**d**) Mg0.6Ca2Ag/PEO after 4 days of immersion in SBF.

Some researchers of Mg–Ca systems have reported that for Ca content below 1.25% the corrosion process is driven from a general mechanism to localized corrosion due to severe electrochemical activity [18]. Other studies suggest that when the amount of Ag in MgAg alloys is increased [16] (1.15 mm/year for Mg2Ag to 1.43 mm/year for Mg6Ag) the Ag-containing second phase does not play an important role in microgalvanic activities, as the responsibility for these phenomena fall to the impurities [48,49]. However, in our case the corrosion rate was 7.78 mm/year for Mg0.6Ca2Ag and 3.64 mm/year for Mg0.6Ca2Ag/PEO. The elevated degradation rate was observed in the present study for Mg0.6Ca in comparison with other Mg–Ca systems [25] and especially for Mg0.6Ca2Ag.

It is evident that, in both alloys, intermetallic particles disclosed some regions with active behaviour that depended on the Ca/Ag ratio in the region (Figures 4 and 5). Such regions present galvanic micro-couples with a minimum anodic surface compared to a large cathodic surface of α-Mg causing dissolution of the intermetallic particles. It can be clearly appreciated from Figure 1a,c that in the Mg0.6Ca2Ag alloy the grain boundary network is much more decorated with intermetallic particles than in Mg0.6Ca. Consequently, dissolution of the grain boundaries can lead to an easy fall-out and loss of the cathodic α-Mg grains or clusters of grains (Figure 12). This anodic behavior of the particles may explain the extremely fast degradation rate of the Mg0.6Ca2Ag alloy.

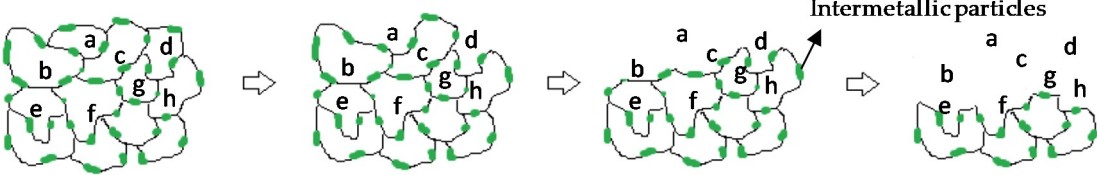

**Figure 12.** Schematic diagram of corrosion attack at the grain boundary and loss of grains (labeled with letters).

### 3.3. Fluoride Release

Figure 13 shows progressive $F^-$ ion release from Mg0.6Ca/PEO during 24 h of immersion in 0.9% NaCl. After 24 h of immersion, 33.757 μg/cm$^2$ of $F^-$ was released, which corresponds to 1.35 ppm

or 71 μM of $F^-$ for the volume of the immersion solution. According to the solubility products, $K_{sp}$, of fluoride containing crystalline phases in the coating, $MgF_2$, $CaF_2$, and $Ca_5(PO_4)_3F$ ($5.16 \times 10^{-11}$, $3.58 \times 10^{-11}$, and $8.6 \times 10^{-61}$, respectively [50]), the dissolution of these compounds will produce up to a maximum of 0.88 mM of free $F^-$. Hence, it would be reasonable to expect a further increase of $F^-$ with time until a solubility limit is reached. In our previous work, it was demonstrated that this limit can be reached in 12 weeks [33]. For comparison, according to The World Health Organization (WHO), fluoride content in drinking water should not exceed 1.22 mg/L (~1 ppm), the optimal daily fluoride consumption for an adult is between 1.4–3.4 mg·day$^{-1}$, whereas the average fluorine concentration in the human blood plasma is 19 ppb [51]. Therefore, the fluoride liberation from Mg0.6Ca/PEO in a 24-h period appears to be safely within these guidelines without a danger of intoxication. Further studies are needed in order to evaluate the potential antibacterial effect of the fluoride in the coating.

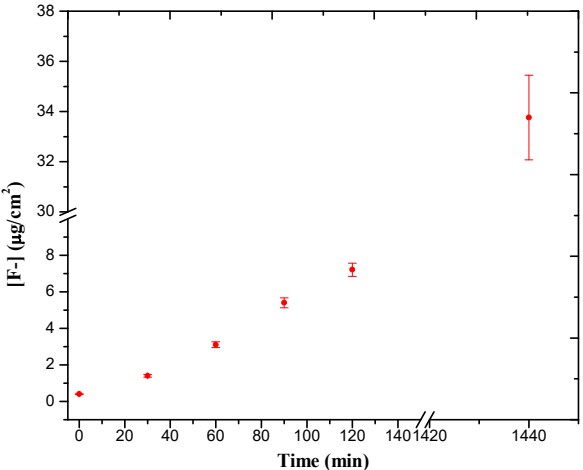

**Figure 13.** Fluoride ions released of PEO coating for 120 min.

## 4. Conclusions

- PEO coatings of 32–35 μm thickness, containing bioactive calcium fluoride, magnesium fluoride and fluorapatite phases, were generated on Mg0.6Ca and Mg0.6Ca2Ag alloys for biomedical implant applications.
- The hydrogen evolution rate of PEO-coated Mg0.6Ca 3D prototypes evaluated during long-term immersion tests (up to 60 days), was 3.86 mL/cm$^2$ week during the first month of immersion in SBF and 1.75 mL/cm$^2$ week during the second month of immersion. The uncoated alloy generated two times more hydrogen in the first month of the test.
- The degradation mechanism during immersion of the studied materials corresponds to a generalized corrosion in both alloys. The Mg0.6Ca2Ag alloys degrade much faster than Mg0.6Ca due to the varied content of Ag and impurities in Mg–Ca–Ag intermetallic particles, which are mainly distributed along the grain boundaries; some of them acting as local anodes and causing the loss of the whole grain or clusters of grains. This also happens in Mg0.6Ca but to a much lesser extent, which leads to a slower corrosion process.
- The PEO coatings liberate fluoride ions during immersion in 0.9 M NaCl. The fluoride release is increased throughout 24 h of immersion time for Mg0.6Ca/PEO. The fluoride reserves of the coating have not been completely depleted in the course of the 24 h immersion.

**Author Contributions:** Formal Analysis, L.M., M.M., B.M., R.A. and E.M.; Funding Acquisition, R.A. and E.M.; Investigation and Methodology L.M. and M.M.; Resources, M.M. and E.M.; Supervision, E.M. and R.A.; Writing—Original Draft Preparation, L.M., M.M. and B.M.; Writing—Review & Editing, E.M. and R.A.; Conceptualization, L.M., M.M., B.M., R.A. and E.M.; Software, L.M., M.M., B.M., R.A. and E.M.; Validation, L.M., M.M., B.M., R.A. and E.M.; Data Curation, L.M., M.M. and B.M.; Visualization, L.M., M.M., B.M., R.A. and E.M.; Project Administration, R.A. and E.M.

**Funding:** This work was partially supported by (MAT2015-73355-JIN) and ADITIMAT-CM (S2018/NMT-4411). M.M. is grateful the Ramon y Cajal Programme (MICINN, Spain, RYC-2017-21843).

**Acknowledgments:** The authors would like to acknowledge the Magnesium Innovation Centre (MagIC, Helmholtz-Zentrum Geesthacht) for supplying the alloys.

**Conflicts of Interest:** The authors declare no conflict of interest.

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
