# Peer review of "Degradation Behaviour of Mg0.6Ca and Mg0.6Ca2Ag Alloys with Bioactive Plasma Electrolytic Oxidation Coatings"

_coatings, doi:10.3390/coatings9060383_

Reviewer 1 Report

Review to the article ” Degradation behaviour of Mg0.6Ca and Mg0.6Ca2Ag alloys with Bioactive Plasma Electrolytic Oxidation (PEO) Coatings” submitted to Coatings MDPI for publication.

The work investigates syntheses of protective coatings on MgCa substrate using Plasma electrolytic oxidation in aqueous electrolyte. The coated and non-coated samples were characterized by

 SEM, TEM, EDX, SKPFM, XRD. Corrosion rate was determined by monitoring of hydrogen release during corrosion. The coating decreased corrosion rate that makes possible to use material for implants preparation for the man body.

In general, the work is done on good scientific and engineering level and can be published after major revision.

Critical comments.

Title. “bioactive coatings”. Probably the materials under the scope have to be not active but passive (from corrosion point of view) and haven’t bio activity and influence on the biological processes in our body. Probably it is better replace “bioactive” for something like: “material -implant application”.

In introduction is good to say about alloying of Mg (e.g. Al, Zn) with aim to increase the corrosion stability. In opposite heavy metals Fe and Ni in the technical and polluted alloys accelerate corrosion significantly because they works as a cathodes for water reduction that accelerate de-passivation of Mg braking down passive MgO film.  It leads to interaction of active Mg surface with water and effective hydrogen release. It makes sense because Ag is effective cathode for water reduction and accelerates Mg corrosion, thus it is not surprise.

Please, find these data in the literature and briefly discuss them in introduction.

2. Please, give literature references on the PEO protocols.

Please, give the composition of m-SBF electrolyte.

Page 3, “SKPFM) with Pt coated Si tip operated in tapping mode”. I don’t think that SKP is measured in taping mode. You have to separate probe from the sample to get the difference of Volta potentials.

“was changed every 48 h or as needed if pH>8.4”. It is natural that Mg will alkalize the solution to pH 10.5. It partially decreases the corrosion rate. Buffering pH 8.4 will accelerate corrosion significantly relatively free (non-controlled) conditions. I thing pH 10.5 will be in our body also. Thus, I don’t see the point to exchange the solution and keeping pH 8.4. It will accelerate corrosion. Please add to article your explanation.

Error! Reference source not found” It is instead of the reference to Figure. It makes difficult the understanding; you have to remove these things.

Ag creates local cathodes which have to increase Volta potential relatively Mg matrix. If we will measure separated pure metal phases, the difference in potentials (measure applying SKP) will be more than 1 V.

“Mg 2Ca shows an anodic behaviour” it is not behaviour, it is more negative potential and the location can be anode in case, if corrosion starts. Behaviour it is process, if no corrosion it is just probable property.

Figure 4, what it is impurity, it is constituent.

“α-Mg matrix (~50 mV). It is” It is better to add delta D E, the difference in potentials between matrix and particular location. Anodic location will be negative with”-” and cathodic “+”.

The mistake in Figures numbering, you have two Figures 4.

“cathodic behaviour (~20mV) in” . Again behaviour, at present the system is not corroding, no any cathodic or anodic behaviour (processes). Add “+” before 20 mV.

Figure 5, You have Ag in negative and positive locations. Are you Sure, about negative location, it means Ag anode. It is impossible.

“whereas the low content of Ag in the inclusion leads to an anodic performance with ”

Ag will be always cathode in this system even in low potential location. Ag was not found in the coating because it doesn’t oxidize.

“The decrease of hydrogen evolution rate is due to the generation of corrosion products layer which acts as a protective coating.” It is difficult to think that corrosion products decrease water reduction rate, probably they buffer near surface pH bigger than 8.4.

“may be associated with the anodic behaviour of the grain boundaries as disclosed”.

In Figures 4 and 5, I did not find grain boundaries and that they obtain more negative potential. You have to show it exactly or say more carefully. In fact, Mg corrodes significantly by disintegration and small metal particles (bigger than grains dimension) are removed from the surface. It is result of non-uniform dissolution or hydrogenation with formation of surface MgH2. No data about anodic boundaries. Thus Figure 32.??? Can be wrong. Can you show any proves? Is it valid for pure Mg (grains disintegration)?

Conclusion

“Mg-Ca-Ag intermetallic particles, which are mainly distributed along the grain boundaries and act as local anodes”. It is needed in additional proves.

Author Response

We would like to thank the reviewer for his/her positive comments and his/her suggestions to improve our work. The modifications are highlighted in the revised Manuscript and the response for his/her comments is detailed in the documents “Comments Reviewers 1”.

Reviewer 2 Report

Abbreviation (PEO) should not appear in the title Abstract: plasma electrolytic oxidation should be abbreviated as  (PEO), not vice versa Page 3: the spectres -> the spectra Fig. 4 outher -> outer, intermedia -> intermediate Fig. 13 provide a fit and error bars Fig. 12 Show the grains to be lost Why section 6 is Patents, and no patent information appears there

Author Response

We would like to thank the reviewer for his/her positive comments and his/her suggestions to improve our work. The modifications are highlighted in the revised Manuscript and the response for his/her comments is detailed in the documents “Comments Reviewers 2”.

Reviewer 3 Report

The manuscript on the "Degradation behaviour of Mg0.6Ca and Mg0.6Ca2Ag alloys with Bioactive Plasma Electrolytic Oxidation (PEO) Coatings" is about the bio active coatings on Mg alloys. While the manuscript is very concise and presented very well, there are a few concerns given below that need to be addressed before considering the manuscript for publication.

The authors talk about only one new material. i.e. Mg0.6Ca2Ag. The results obtained by the authors are not promising for this material. What is the rationale behind selection of this material?

Since the corrosion rates are high for this material, have the authors tried alternative routes to enhance it's resistance? i.e. by changing processing parameters during coating or other such routes?

The results presented by the authors shows no effect of PEO coating on this alloy. So, what is the novelty and significance of this work?

There are multiple grammatical errors throughout the manuscript. They need to be corrected.

Author Response

We would like to thank the reviewer for his/her positive comments and his/her suggestions to improve our work. The modifications are highlighted in the revised Manuscript and the response for his/her comments is detailed in the documents “Comments Reviewers 3”.

Reviewer 4 Report

Review of article entitled: Degradation behaviour of Mg0.6Ca and Mg0.6Ca2Ag alloys with Bioactive Plasma Electrolytic Oxidation (PEO) Coatings Coatings-516157

Comments for Authors:

The article entitled Degradation behaviour of Mg0.6Ca and Mg0.6Ca2Ag alloys with Bioactive Plasma Electrolytic Oxidation (PEO) Coatings is well written and I do not have special remarks according to the text, however I would suggest add some data about the mechanical properties of coatings, if the future application is strictly connected to the implants technology. If the authors do not have access to mechanical studies equipment, the information about the further studies in this area should be added. Coatings can be the best from chemical and biological point of view, but if they are not mechanically resistant and if they are susceptible to scratching they will not meet the basic requirements for surface-modified implants.

Author Response

We would like to thank the reviewer for his/her positive comments and his/her suggestions to improve our work. The modifications are highlighted in the revised Manuscript and the response for his/her comments is detailed in the documents “Comments Reviewers 4”.

Round  2

Reviewer 1 Report

Thank you.

Reviewer 2 Report

Accept

Reviewer 3 Report

The authors have responded to the comments. The manuscript in its present form can be accepted.